# In Vitro/In Vivo Translation of Synergistic Combination of MDM2 and MEK Inhibitors in Melanoma Using PBPK/PD Modelling: Part I

**DOI:** 10.3390/ijms232112984

**Published:** 2022-10-26

**Authors:** Jakub Witkowski, Sebastian Polak, Zbigniew Rogulski, Dariusz Pawelec

**Affiliations:** 1Faculty of Chemistry, University of Warsaw, Pasteura 1, 02-093 Warsaw, Poland; 2Adamed Pharma S.A., Adamkiewicza 6a, 05-152 Czosnów, Poland; 3Faculty of Pharmacy, Jagiellonian University, Medyczna 9, 30-688 Kraków, Poland; 4Simcyp Division, Certara UK Limited, Level 2-Acero, 1 Concourse Way, Sheffield S1 2BJ, UK

**Keywords:** anticancer drugs, preclinical study, pharmacokinetics, pharmacodynamics, drug combination, PBPK/PD modelling, MDM2 inhibitor, MEK inhibitor

## Abstract

Translation of the synergy between the Siremadlin (MDM2 inhibitor) and Trametinib (MEK inhibitor) combination observed in vitro into in vivo synergistic efficacy in melanoma requires estimation of the interaction between these molecules at the pharmacokinetic (PK) and pharmacodynamic (PD) levels. The cytotoxicity of the Siremadlin and Trametinib combination was evaluated in vitro in melanoma A375 cells with MTS and RealTime-Glo assays. Analysis of the drug combination matrix was performed using Synergy and Synergyfinder packages. Calculated drug interaction metrics showed high synergy between Siremadlin and Trametinib: 23.12%, or a 7.48% increase of combined drug efficacy (concentration-independent parameter β from Synergy package analysis and concentration-dependent δ parameter from Synergyfinder analysis, respectively). In order to select the optimal PD interaction parameter which may translate observed in vitro synergy metrics into the in vivo setting, further PK/PD studies on cancer xenograft animal models coupled with PBPK/PD modelling are needed.

## 1. Introduction

One of the first targeted therapies approved was a drug combination targeting the MAPK (mitogen-activated protein kinase family) signaling pathway of Dabrafenib (BRAF inhibitor) and Trametinib (MEK inhibitor) [1]. The Dabrafenib and Trametinib combination showed synergistic efficacy and significantly increased the overall survival of melanoma patients. Preclinical evidence suggests that the drug combination with MEK and MDM2 (mouse double minute 2) inhibitors may also act synergistically in the treatment of melanoma [2]. This pharmacodynamic interaction has been characterized at a molecular level, and may be explained by the DUSP6 mechanism (DUSP6 suppression followed by increased p53 phosphorylation) in BRAFV600E and p53WT melanoma cells, which leads to synergistic induction of the expression of genes encoding PUMA and BIM that increase apoptosis ratio and growth inhibition of melanoma cells [3]. There is also in vivo evidence in animals with melanoma tumour xenografts that this synergistic efficacy may be efficient in the treatment of skin cancer [4]. However, long-term administration of MDM2 and MEK inhibitors can lead to acquired resistance caused by the mechanism of spontaneous p53 and MAP2K1 (MEK1) mutations or expression of BRAF-V600E splice variants [5,6,7]. Thus, combining these two classes of drugs can bring benefit to the patients by restoring anticancer activity (salvage therapy if one acquires resistance against either drug [8]) or delaying development of resistance to the treatment. It should be noted that combining the drugs may increase the adverse effects observed in patients. Therefore, the selection of drugs with a different profile of adverse effects for drug combination is very important for the sake of patient safety. Examination of this particular drug combination on healthy cells was not in the scope of this publication, but previously published data for MDM2 and MEK inhibitors indicate very low toxicity in healthy cells [9,10]. Moreover, since such a drug combination is currently the subject of many studies in clinical trials (ClinicalTrials.gov identifiers: NCT02110355, NCT03714958, NCT02016729, NCT01985191, NCT03566485) it is assumed that this combination is generally safe and not excessively toxic to healthy cells. This drug combination utility was confirmed in the clinical setting with moderately active MDM2 inhibitor AMG232 [11], but it is believed that the next generation of more potent MDM2 inhibitors, such as Siremadlin (HDM201), may further enhance this synergistic drug interaction.

In order to assess how the addition of Siremadlin to Trametinib could improve anticancer response, performance of preclinical translational studies and the development of in vitro/in vivo translational methods are truly essential. A bench-to-bedside approach for drug combination may be possible only when PK/PD data for both drugs are available because it must account for the interaction between two (or even more) drugs at two different levels: pharmacokinetic (PK) and pharmacodynamic (PD).

Analyses of the interactions between two or more drugs at the PD level are impeded by a lack of consensus on which method/theoretical model should be used to describe drug interaction. The quantification of the interaction between drugs is based on the comparison of the observed combination response to the expected effect predicted by a reference model under the assumption of non-interaction of those drugs. Depending on whether the combination response is greater or less than what is expected, the drug combinations can be classified as synergistic or antagonistic, respectively. In a case where the drug combination response equals the expected effect, it can be classified as additive (for some authors, such lack of drug interaction is also referred to as independence or noninteraction [12]). Historically, there were also other types of drug interactions, such as inertism or coalism, described by Greco et al., and Roell et al. [12,13]. Over the past years, many drug interaction frameworks have been developed, including the most recognized models, the Loewe additivity [14], Bliss independence [15], or highest single agent (HSA) [16] models, and the most recent but less recognized models that overcome many limitations of already existing models, such as zero interaction potency (ZIP) [17] or multi-dimensional synergy of combinations (MuSyC) [18].

Choosing the model to evaluate combination data became very problematic in the light of scientific discussion over the past years. There has often been a dilemma when a drug combination is classified as synergistic according to one model but antagonistic by the other [16,19,20]. According to Tang et al. [19], this phenomenon can be explained by consistency between models, which can be indicative of the degree of drug interaction; e.g., if both the Bliss model and the Loewe model classify a drug combination as synergistic, then it may be described as a strong synergy. On the other hand, if the drug combination interaction is classified as synergistic according to one model only, then it may be described as a weak synergy or additivity. Synergy and efficacy concepts are highly related, but it is very important to not treat them as the same. Synergy is a type of drug interaction and a measure of its degree, while efficacy is the magnitude of the phenotypic response of a drug combination. It can be observed that a combination of drugs can be highly synergistic, whereas its response may not be sufficient to achieve therapeutic efficacy. It is also possible that a drug combination can show a strong response which is not related to a synergistic interaction (e.g., only one drug is responsible for the observed response) [18,19,21].

Unlike the models mentioned above, the MuSyC model addresses those two concepts of drug interaction by describing their interaction with three different metrics that decouple synergy or antagonism based on increased/decreased efficacy (parameter β), potency (parameter α), and cooperativity (parameter γ), which may be advantageous due to disease type. Moreover, those drug interaction parameters are dose-independent and, thus, observed synergistic interaction may not lead to ambiguous results (for a given combination drug interaction may be synergistic or not, as both drugs can synergize at some concentrations, and antagonize at others) [18]. 

Due to the nature of the drug combination studies which are typically conducted in the high throughput screening (HTS) format, the most practical solution would be to use software capable of calculating multiple drug interaction metrics for large datasets. Unfortunately, there are only a few software packages for which such features are available: Synergyfinder [22], Synergy [23], and Combenefit software [24]. Because the Combenefit software has not been developed since 2016 and provides analysis with the same models as Synergyfinder (Loewe, Bliss and HSA), analysis involving Combenefit was not considered in this work. 

The Synergyfinder package allows for calculating delta score (δ) value (which corresponds to the percentage of drug combination response beyond expectation) for Loewe, Bliss, HSA and ZIP models. For example, a delta of 10 would indicate that the drug combination will produce, on average, 10% more response compared to the expected effect predicted by given theoretical drug interaction model (Loewe, Bliss, HSA and ZIP models), which we would refer here as synergistic drugs interaction, while a delta of −10 would indicate an antagonistic drug interaction with the same level of magnitude in this case. Applying a threshold of 5% response (delta score δ = 5), which is the typical noise level in large-scale drug combination experiments, minimizes the rate of false-positive results [17,25]. Therefore, in this context, and according to Tang’s group’s experience, classification of drug combinations was formulated based on delta score value, as shown in Table 1 [26].

Despite the fact that the Synergy package permits use of several drug interaction models, such as concentration (dose)-independent parametric models (MuSyC, Zimmer, and BRAID) and concentration (dose)-dependent nonparametric models (Loewe, Bliss, HSA, CombinationIndex, Schindler, and ZIP), its main focus was on the MuSyC model to determine which drug interaction metric among the efficacy (parameter β), potency (parameter α), and cooperativity (parameter γ) parameters would be the most significant in the context of in vitro/in vivo translation of the drug combination interaction at the pharmacodynamics level. Parameter β may be interpreted as the percent increase in maximal efficacy of the combination over the most efficacious single agent. Parameter α quantifies, in fold, how the effectiveness of one drug is altered by the presence of the other (fold change in the potency of combined drugs). Gamma parameter provides information about the change of a drug’s Hill slope (cooperativity) due to the other drug. There are two values for *α* and *γ* because each drug can independently modulate the potency and cooperativity of the other. Classification of drug combination interaction in the MuSyC model based on α, β, and *γ* score values is shown in Table 2 [27,28]. 

Translation of the in vitro into in vivo synergistic efficacy must cover interactions at the PK and PD levels. One of the approaches allowing for the incorporation of such interactions is physiologically based pharmacokinetic/pharmacodynamic (PBPK/PD) modelling.

The main goal of this work is focused on estimation of the PD interaction parameter, which may serve for translatability of in vitro drug combination results into in vivo settings. Such an approach, involving results from in vitro cytotoxicity studies coupled with PBPK/PD modelling, may facilitate the determination of the most synergistic and efficacious schedule and dose levels for Siremadlin and Trametinib in mice, in vivo, and also may be the basis for better estimation of drug combination efficacy in melanoma patients.

## 2. Results

### 2.1. In Vitro Cytotoxicity

Siremadlin and Trametinib efficacy in monotherapy and in combination was studied in in vitro A375 human melanoma cells with the use of an MTS assay. One of the limitations of the preclinical drug combinations is the reproducibility of the measured drug interaction metrics [29,30,31]; thus, additional in vitro efficacy study was performed with the use of RealTime-Glo assay to confirm the efficacy and observed drug interaction metrics. Results indicate that both drugs are very active against A375 melanoma cells (see Table 3).

Despite the high efficacy of tested compounds, some populations of cells remained resistant even at high concentrations (see Appendix A). RealTime-Glo assay revealed that the killing effect and its intensity for both drugs was concentration- and time-dependent (see Figure 1 and Figure 2).

The combination of those two compounds caused a profound increase in cytotoxicity, demonstrating an increase in potency and efficacy against A375 cells. For survival curve-shift, see Figure 3 and Figure 4.

### 2.2. In Vitro Drug Combination Analysis

Results from both assay methods (MTS and RealTime-Glo) with analysis involving use of Synergyfinder package (δ score from ZIP, Loewe, has, and Bliss models) were generally consistent, comparable, and indicated a synergistic interaction between studied drugs. Two exceptions were the ZIP model (RealTime-Glo assay) and Bliss model (MTS assay); however, calculated δ scores were very close to the synergistic threshold. Synergistic delta score is also reflected in calculated mean across the methods and models (see Table 4 and Appendix A).

Results from both assay methods with analysis using the Synergy package (MuSyC model) were comparable for α_21,_ β, and *γ*_21_ parameters (especially in the 48–80 h interval for α_21_ and *γ*_21_ parameters, as shown in Appendix A), indicating synergistic interaction in terms of increased potency (α parameter) and efficacy (β parameter) between Siremadlin and Trametinib, but not in terms of increased cooperativity (γ parameter); however, it seems that synergistic cooperativity is more important in neurological disorders than in treating cancer; thus, lack of synergy in this metric is clinically not relevant [22] (see Table 5 and Appendix A).

### 2.3. Siremadlin and Trametinib Pharmacokinetics (PK)

Pharmacokinetic profiles were determined after single oral administration of HDM201 (100 mg/kg) and Trametinib (1 mg/kg) in vehicle formulation in CD-1 nude mice. Initial analysis of compound concentrations included plasma and A375 tumour tissue homogenates. Additionally, analysis included HDM201 and Trametinib administered in combination (100 + 1 mg/kg, respectively) to determine whether there were interactions at the PK level, as shown in Figure 5, Figure 6, Figure 7 and Figure 8.

The values of calculated PK parameters suggest that HDM201 and Trametinib are absorbed relatively quickly (typical Tmax are in the 1.5–4 h range) and maintain high exposure in plasma within 24 h. Data from A375 tumour tissue indicate that both compounds are well distributed in the tumour achieving higher maximal concentrations and exposure (Cmax and AUC, respectively) as shown in Table 6. PK analysis revealed that plasma and tumour Cmax and AUC for HDM201 have higher values when co-administered with Trametinib, while for Trametinib those parameters are higher only in A375 tumour. Interestingly, in plasma, Cmax and AUC for Trametinib are significantly lower when co-administered with HDM201, as depicted in Figure 6.

### 2.4. Siremadlin and Trametinib Pharmacodynamics (PD)

The studied compounds, namely HDM201 and Trametinib, were tested separately and in combination as a therapy against an A375 melanoma tumour model. Tested compounds were administrated in three and six doses per schedule. All tested compounds decreased tumour volume in comparison to the group treated by formulation. Data from this efficacy study indicate that HDM201 and Trametinib are more efficacious when they are used in combination, compared to their efficacy when administered separately (please refer to maximal tumour growth inhibition (TGI) percentage, presented in Table 7, and compounds efficacy in single and combined administrations, shown in Figure 9).

## 3. Discussion

In vitro cytotoxicity data demonstrated high efficacy of Siremadlin and Trametinib against A375 cells, which justifies further studies in in vivo models. Calculated Trametinib IC50 values were very similar to the values reported previously in the literature [32,33]; however, greater difference was observed for Siremadlin IC50. Differences in IC50 values may be explained by dissimilarities in methods used for counting living cells (different assays) or by the influence of cell-seeding density on cytotoxic sensitivity [34]. Performed experiments revealed that cytotoxicity of the studied compounds is concentration- and time-dependent with an initial delay of response. The delay in response to these drugs is most likely related to the duration of signal transduction cascade associated with the activation of the p53-MDM2 and MAPK pathways, resulting in cell death.

Resistance is an inherent part of anticancer treatment; therefore, the population of resistant cells was assessed for both drugs in the performed study, the description of which may play a critical role in predicting and optimizing treatment response and may improve therapy scheduling [35,36]. However, further in vitro studies on drug-resistant A375 melanoma sublines with the Siremadlin and Trametinib combination would be needed for in-depth analysis of resistance mechanisms, and to test if such combination treatment would be suitable for prolonged treatment, which is often characterized by increased resistance [7,37,38,39,40,41,42]. Results from drug interaction analysis indicated synergistic interaction between the drugs studied in an A375 melanoma model. The use of two different cytotoxicity assays (MTS and RealTime-Glo) indicated consistency in the drug interaction metrics obtained. Regarding analysis at particular timepoints, it can be noted that in analysis with use of the MuSyC model, the values of the α_21_ and *γ*_21_ parameters up to 36 h were very high and irregular. This may be caused by different initial responses of cells, since therapy started when cells were at different stages of the cell cycle. It can be hypothesized that those parameters began to stabilize when the majority of those cells reached the stasis/apoptosis stage.

Synergistic interaction was identified by using a number of the most commonly used theoretical drug interaction models with the use of Synergyfinder and Synergy software. Calculated synergy metrics were different in terms of their magnitude and foundations: a 7.48% increase in drug combination efficacy in the case of Synergyfinder (for the mean concentration-dependent δ parameter, see Table 4) and a 23.12% of increase in efficacy in the case of the Synergy software (for the mean concentration-independent β parameter, see Table 5). Those calculated synergy metrics will serve as a PD interaction parameter for further translational PBPK/PD modelling. Additional information obtained from this in vitro study about the delay of the response and the percentage of resistant cells, in terms of the total cell population, is very useful and will be also incorporated into PD model development.

Initial pharmacokinetic analysis performed on mice plasma indicated that HDM201 and Trametinib undergo fast absorption, as the Tmax values were in the 1.5–4 h range; however, due to a limited quantity of available timepoints (sparse sampling), these values may be not accurately determined. HDM201 and Trametinib in plasma and A375 tumours are characterized by high exposure within 24 h.

The pharmacokinetic profile of HDM201 co-administered with Trametinib in plasma is characterized by higher maximal concentration and exposure (Cmax and AUC, respectively) than after single drug administration. Interestingly, in the case of Trametinib co-administered with HDM201, the situation is the opposite. Observed Cmax and AUC are lower than after Trametinib administration alone. This may be related to the occurrence of PK interaction; however, further PK studies combined with PBPK modelling are required to prove the existence of this interaction and the mechanism of its formation. Higher Cmax and AUC in A375 tumour tissue than in plasma were observed for both drugs. Such observation is very favourably in context of potential combination therapy using HDM201 and Trametinib. Nevertheless, further PK studies combined with PBPK modelling are required to explain the mechanisms of pharmacokinetic interactions in the tumour compartment. More detailed pharmacokinetic analysis on heart, liver, spleen, muscle, brain, kidney, lung, gut, and skin tissue homogenates, combined with PBPK modelling and simulation, is the main topic of the second part (part II) of this publication cycle [43].

Efficacy data showed that all tested compounds decreased tumour volume in comparison to the group treated by vehicle formulation. Results from this study indicated that HDM201 and Trametinib were much more efficient when they are used in combination, compared to their efficacy when the those two compounds are administered separately. This observation should be supported by the results of subsequent in vivo studies using a higher number of animals per group. Moreover, it remains unclear how the PK interaction combined with PD interaction influences the observed anticancer efficacy for this drug combination. Therefore, further PK/PD studies on mice, coupled with PBPK/PD modelling, are needed in order to determine the mechanism of the PK interaction formation and to select the optimal PD interaction parameter which will translate observed in vitro synergy metrics into in vivo settings. Such an approach may facilitate the determination of the most synergistic and efficacious schedule and dose levels for Siremadlin and Trametinib in in vivo models, and may provide a basis for better estimation of drug combination efficacy in melanoma patients.

## 4. Materials and Methods

### 4.1. Materials

HDM201 (catalog number HY-18658) and Trametinib (catalog number HY-10999) used in this study were obtained from MedChemExpress. RealTime-Glo™ MT Cell Viability Assay kit and CellTiter 96^®^ AQueous Non-Radioactive Cell Proliferation Assay (MTS) were provided by Promega. The A375 cell line used in the in vitro studies was obtained from American Type Culture Collection (CRL-1619). PEG 400 (catalog number 81172) and Cremophor RH40 (catalog number 07076) were provided by Merck (formerly Sigma-Aldrich), EtOH (catalog number 1016/12/19) was provided by POCH, and Labrafil M1944CS (catalog number 178290) was provided by Gattefosse. For drug combination in vivo studies, A375 cell line was provided by European Collection of Authenticated Cell Cultures (88113005).

### 4.2. Software

For in vitro drug combination studies, raw data processing was performed in Microsoft Excel 2016 with the use of Excel Visual Basic for Applications (VBA) macros. PK parameters and TGI values were estimated with Microsoft Excel (Excel version 2016, Microsoft Corporation, Redmond, WA, USA, 2016, https://www.office.com). For in vitro drug combination analysis, Synergyfinder (version 2.5.1 compiled on 01.12.2020 from Github resource https://github.com/shuyuzheng/synergyfinder) and Synergy (version 0.4.5 compiled on 13.02.2021 from Github resource https://github.com/djwooten/synergy) packages were used. Visualization of the in vivo efficacy and calculation of the IC_50_ values by curve fitting were performed using GraphPad Prism version 9.3.1 for Windows, GraphPad Software, San Diego, CA, USA, 2022, www.graphpad.com.

### 4.3. In Vitro Drug Combination Studies

Inhibition of tumour cell viability after single and combination drug treatment was measured with the use of MTS assay (CellTiter 96^®^ AQueous Non-Radioactive Cell Proliferation Assay) and RealTime-Glo assay (RealTime-Glo™ MT Cell Viability Assay) in A375 cell line using standard manufacturer protocols. Briefly, cells were plated at optimized seeding density (0.5 × 10^3^ cells/well) in a 96-well culture plate in an appropriate cell culture medium (DMEM 4.5 g/L glucose supplemented with 10% *v*/*v* FBS), cell cultures were stimulated with compounds 24 h after cell seeding with MDM2 and MEK inhibitors. In the MTS assay, compounds were added at 5 concentrations, ranging from 12.5 nmol/L to 200 nmol/L and 0.25 nmol/L to 4 nmol/L (for HDM201 and Trametinib, respectively) along with a dimethyl sulfoxide (DMSO) control. In the RealTime-Glo assay, compounds were added at 7 concentrations, ranging from 62.5 nmol/L to 4000 nmol/L for HDM201 and 0.625 nmol/L to 40 nmol/L for Trametinib, with a dimethyl sulfoxide (DMSO) control. Drug combinations were tested in the matrix layout with the increasing concentrations of both drugs (please see Appendix A). Tumour cells’ viability was measured after 72 h of cell incubation in the presence of tested compounds in the MTS assay and at 0,12, 24, 28, 32, 36, 48, 52, 56, 60, 72, 76, 80 h timepoints in the RealTime-Glo assay (with exception in first experiment, performed with RealTime-Glo assay, on which 28 and 32 h timepoints were not measured). Several independent assay repetitions were performed for MTS (*n* = 4) and RealTime-Glo assays (*n* = 3).

### 4.4. Drug Combination Interaction Analysis

Drug combination interaction analysis was performed on in vitro cytotoxicity data from MTS and RealTime-Glo assays. For RealTime-Glo data, all further calculations of drug interaction parameters were performed in the 28–80 h time range, due to the lack of significant efficacy of single Siremadlin and Trametinib at 12 and 24 h timepoints and the impossibility of finding proper drug interaction model fit (R^2^ < 0.8). Analysis with the use of the Synergyfinder package (version 2.5.1 compiled on 01.12.2020 from Github resource https://github.com/shuyuzheng/synergyfinder) was performed with a script written in the R language in RStudio Version 1.2.5001 Build 1468 (see Appendix A). For Synergyfinder analysis parameters, see Appendix A. For a more detailed description of all Synergyfinder functions, please see its documentation and user instructions [44].

Analysis with the use of the Synergy package (version 0.4.5 compiled on 13.02.2021 from Github resource https://github.com/djwooten/synergy was performed with a script written in Python language in Python 3.9.12 (see Appendix A). For Synergy analysis parameters, see Appendix A.

### 4.5. Studies Involving Animals

Crl:CD-1-*Foxn1^nu^* female mice, 4–5 weeks old, from Charles River Germany, inoculated subcutaneously with A375 cells, were used for in vivo studies. Determination of compound concentrations in plasma and A375 tumour tissue homogenates were performed with the use of quantitative LC-MS/MS system. Plasma and tissue samples were resected at the following timepoints: 1.5, 4, 8, 24 h (*n* = 3 per timepoint) after oral administration. Pharmacokinetic parameters AUC, C_max_, and T_max_ were calculated using MS Excel 2016. Area under the concentration versus time curve was calculated using the linear trapezoidal rule [45].

Determination of tumour growth was performed after oral gavage of Vehicle (60% PEG 400 (*v*/*v*), 10% Cremophor RH40 (*v*/*v*), 10% EtOH (*v*/*v*) and 20% Labrafil M1944CS (*v*/*v*)), Siremadlin, Trametinib or their combination in Vehicle. The volume of the administered formulation (10 mL/kg) of the compounds was always adjusted to the mice body weight. Initial tumour volumes, doses, dose schedules, and number of animals in particular in vivo studies are summarized in Table 8.

Tumour volume (V) was recorded with an electronic calliper 2–3 times a week and was calculated based on its length and width, using the prolate ellipsoid equation [46] (Equation (1)):V (mm^3^) = d^2^ × D/2(1)
where d is the tumour width (mm) and D is the tumour length (mm).

Tumour growth inhibition (TGI) value was calculated using Equation (2) [47]:TGI (%) = (100 − (T/C × 100))(2)
where C is the mean tumour size in control group (mm^3^) and T is the mean tumour size in treated group (mm^3^).

## Figures and Tables

**Figure 1 ijms-23-12984-f001:**
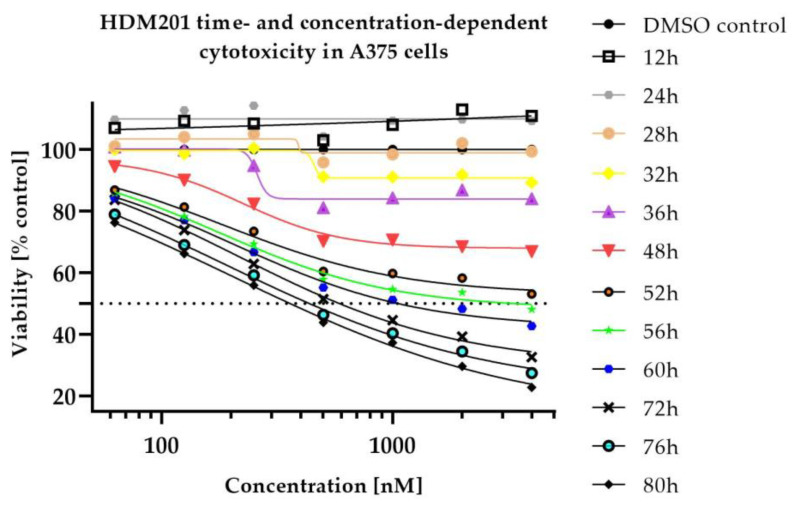
Time- and dose-dependent cytotoxicity of HDM201 in A375 cells. Mean from *n* = 3 (RealTime-Glo assay).

**Figure 2 ijms-23-12984-f002:**
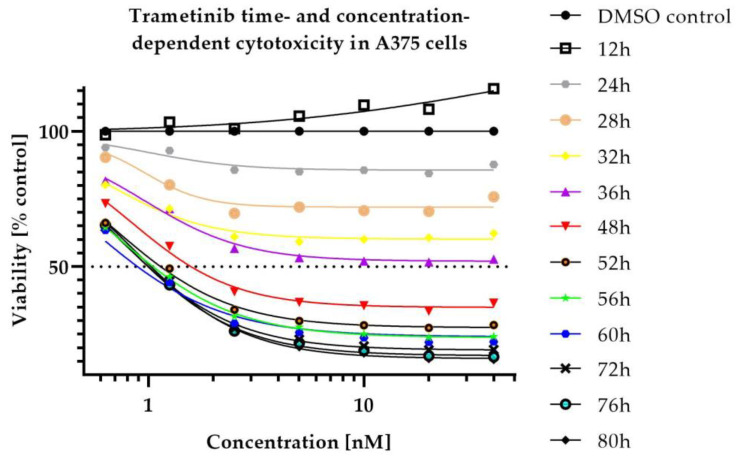
Time and dose-dependent cytotoxicity of Trametinib in A375 cells. Mean from *n* = 3 (RealTime-Glo assay).

**Figure 3 ijms-23-12984-f003:**
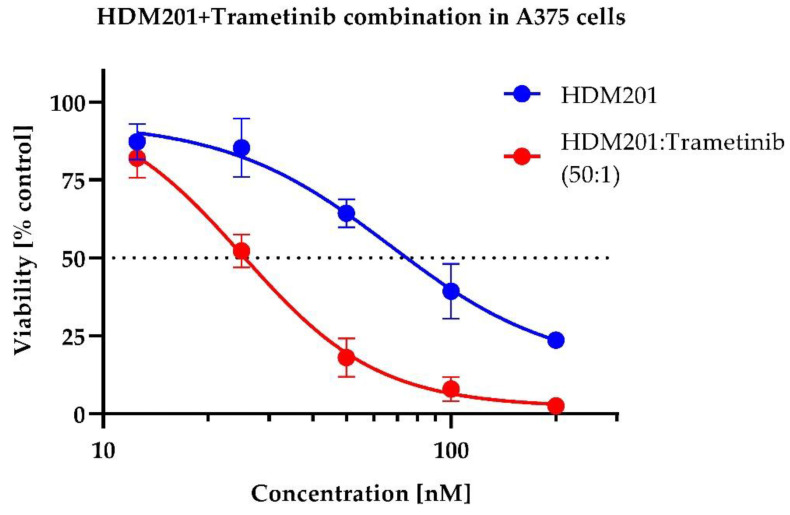
Curve shift for HDM201 combination with Trametinib. MTS assay, mean from *n* = 4.

**Figure 4 ijms-23-12984-f004:**
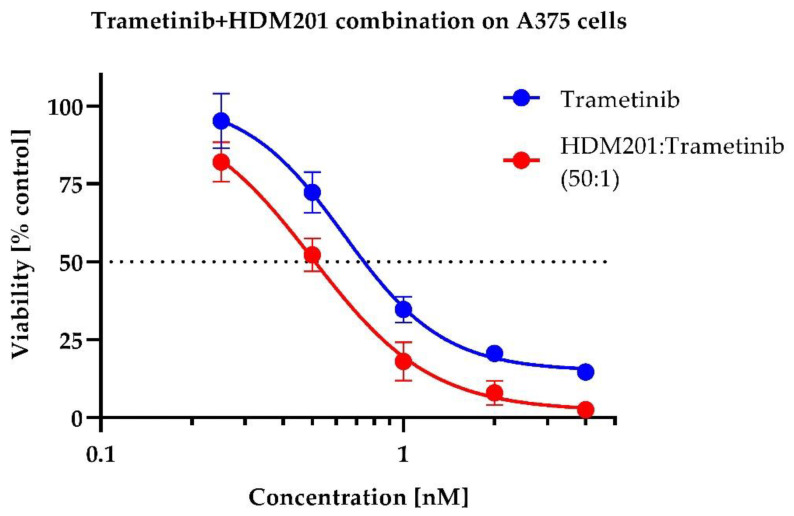
Curve shift for Trametinib combination with HDM201. MTS assay, mean from *n* = 4.

**Figure 5 ijms-23-12984-f005:**
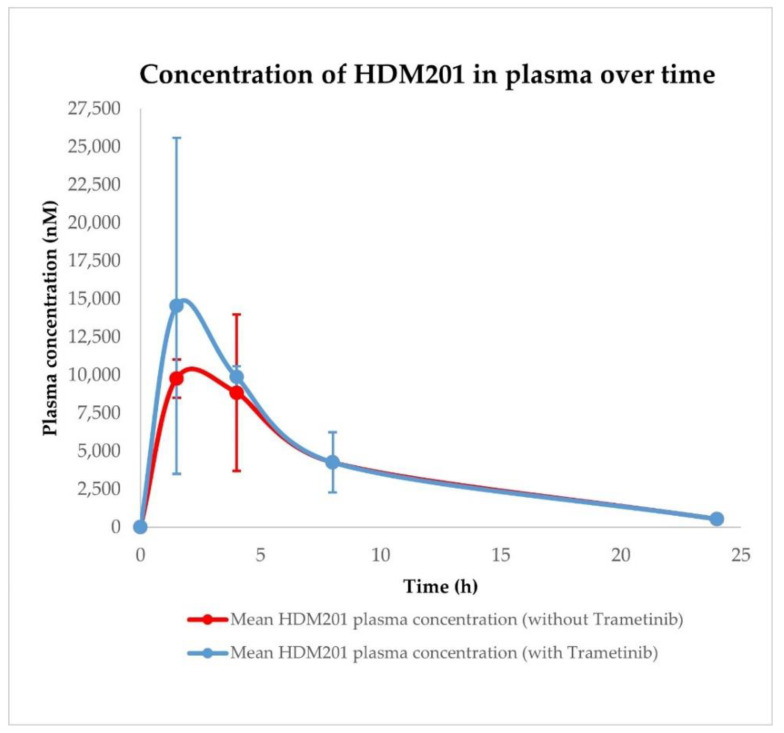
Pharmacokinetic profiles of HDM201 (Siremadlin) in plasma after administration with and without Trametinib. Observed data are means ± standard deviation (SD) from *n* = 3.

**Figure 6 ijms-23-12984-f006:**
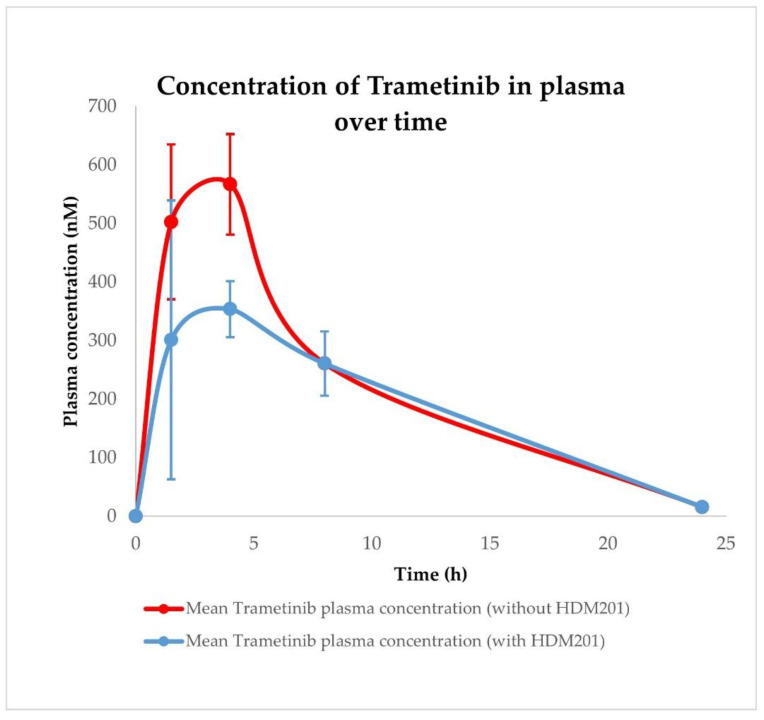
Pharmacokinetic profiles of Trametinib in plasma after administration with and without HDM201. Observed data are means ± standard deviation (SD) from *n* = 3.

**Figure 7 ijms-23-12984-f007:**
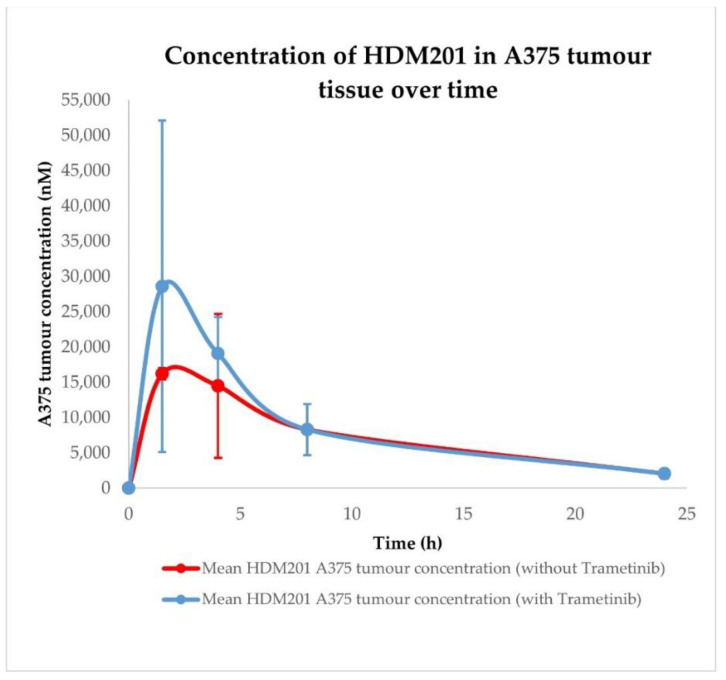
Pharmacokinetic profiles of HDM201 (Siremadlin) in A375 tumour tissue after administration with and without Trametinib. Observed data are means ± standard deviation (SD) from *n* = 3.

**Figure 8 ijms-23-12984-f008:**
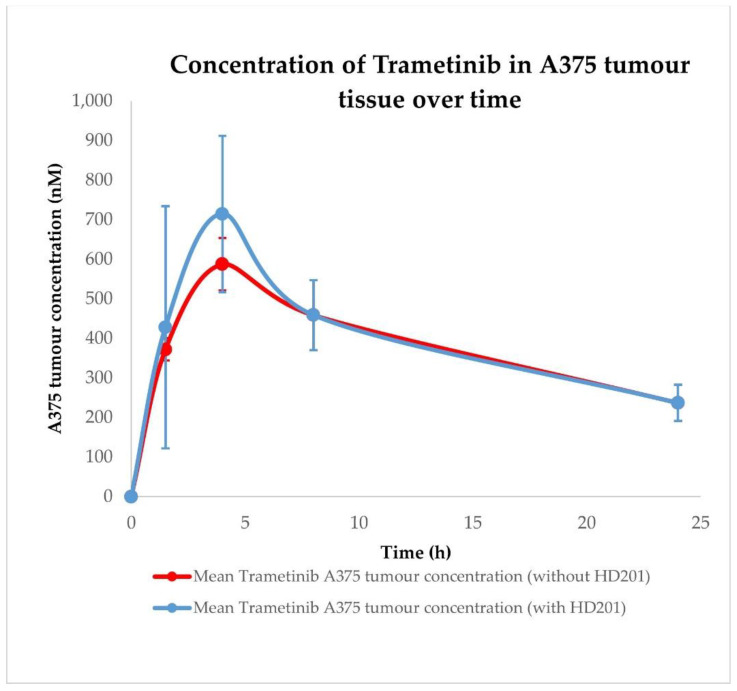
Pharmacokinetic profiles of Trametinib in A375 tumour tissue after administration with and without HDM201. Observed data are means ± standard deviation (SD) from *n* = 3.

**Figure 9 ijms-23-12984-f009:**
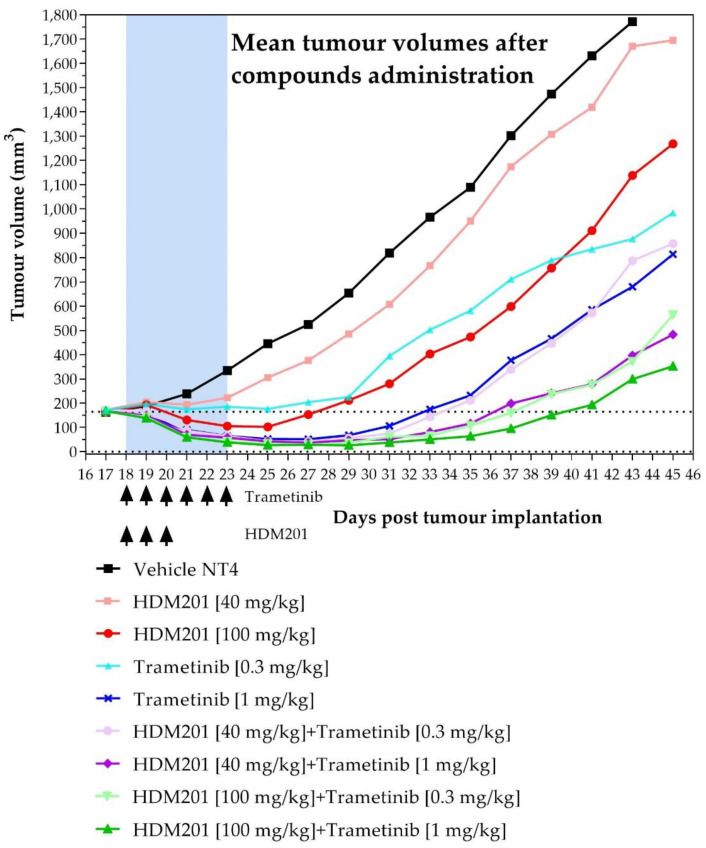
In vivo efficacy of HDM201 and Trametinib administered separately and in combination against a mouse model of human melanoma tumour (A375). HDM201 was dosed in a qdx3 schedule and Trametinib in a qdx6 schedule. Observed data are means from *n* = 6.

**Table 1 ijms-23-12984-t001:** Drug interaction classification based on delta score (Synergyfinder package).

Description	δ Score Value
Antagonism	≤−5
Additivity	(−5; 5)
Synergism	≥5

**Table 2 ijms-23-12984-t002:** Drug interaction classification based on alpha, beta, and gamma scores (Synergy package).

Description	α_12_/α_21_ Score Value	β Score Value	*γ*_12_/*γ*_21_ Score Value
Antagonism	<1	<0	<1
Additivity	1	0	1
Synergism	>1	>0	>1

**Table 3 ijms-23-12984-t003:** Comparison of observed Siremadlin and Trametinib IC50 with data from the literature in A375 cells (72 h incubation). For MTS assay, *n* = 4, and for RealTime-Glo assay, *n* = 3.

Compound	MTS IC50 ± SD (nM)	RealTime-Glo IC50 ± SD (nM)	Literature IC50 (nM)
Siremadlin (HDM201)	65.7 ± 4.7	260.1 ± 170.5	764.1 [32] ^1^
Trametinib	0.58 ± 0.03	0.8 ± 0.4	1.0 [33] ^2^

^1^ Cell count was measured using an amount of ATP (CellTiter-Glo assay). ^2^ Cell count was measured using 4′,6-diamidino-2-phenylindole (DAPI) nuclei staining.

**Table 4 ijms-23-12984-t004:** Drug interaction between Siremadlin and Trametinib in A375 cells. For MTS assay, *n* = 4, and for RealTime-Glo assay, *n* = 3. Synergyfinder package analysis.

Assay	Timepoints (h)	ZIP δ ± SD	Loewe δ ± SD	HSA δ ± SD	Bliss δ ± SD	Mean across Models δ ± SD
MTS	72	5.353 ± 2.613	5.111 ± 1.926	12.394 ± 2.085	4.881 ± 3.117	6.935 ± 2.420
RealTime-Glo	28–80	4.858 ± 1.346	7.113 ± 4.355	13.513 ± 3.111	5.540 ± 1.957	7.756 ± 1.614
Mean	-	5.023 ± 1.768	6.446 ± 3.546	13.140 ± 2.769	5.321 ± 2.344	7.482 ± 1.883

**Table 5 ijms-23-12984-t005:** Drug interaction between Siremadlin and Trametinib in A375 cells. For MTS assay, *n* = 4, and for RealTime-Glo assay, *n* = 3. Synergy package analysis.

Assay	Timepoints (h)	α_12_/α_21_ ± SD	β ± SD	*γ*_12_/*γ*_21_ ± SD
MTS	72	2.229 ± 1.065/ 1.498 ± 0.351	0.217 ± 0.045	0.402 ± 0.102/ 0.710 ± 0.286
RealTime-Glo	28–80	2.095 ± 0.780/ 12,507 ± 26,999	0.244 ± 0.050	0.901 ± 0.136/ 6878 ± 21,748
Mean		2.162 ± 0.923/ 6254 ± 13,500	0.231 ± 0.048	0.652 ± 0.119/ 3440 ± 10,874

**Table 6 ijms-23-12984-t006:** Calculated PK parameters for HDM201 and Trametinib in plasma and A375 tumour tissue.

Conditions	Tissue	AUC_0–24h_ ± SD (nM × h)	Cmax ± SD (nM)	Tmax ± SD (h)
HDM201 without Trametinib	Plasma	95,092.97 ± 34,215.83	9777.67 ± 2976.84	1.50 ± 1.44
HDM201 with Trametinib	Plasma	107,993.98 ± 26,303.00	14,559.95 ± 7433.26	1.50 ± 1.44
Trametinib without HDM201	Plasma	5580.83 ± 566.66	567.02 ± 49.38	4.00 ± 1.44
Trametinib with HDM201	Plasma	4484.99 ± 1171.06	353.65 ± 105.55	4.00 ± 1.44
HDM201 without Trametinib	A375 tumour	179,026.48 ± 65,901.61	16,214.30 ± 5459.78	1.50 ± 1.44
HDM201 with Trametinib	A375 tumour	218,677.07 ± 91,168.31	28,613.74 ± 16,751.20	1.50 ± 1.44
Trametinib without HDM201	A375 tumour	9131.17 ± 1296.84	587.25 ± 66.35	4.00 ± 0.00
Trametinib with HDM201	A375 tumour	9656.67 ± 1393.80	714.53 ± 197.48	4.00 ± 0.00

**Table 7 ijms-23-12984-t007:** Mean tumour growth inhibition (TGI) values with standard error of mean (SEM) as metrics of in vivo efficacy of HDM201, Trametinib, and their combination in A375-inoculated CD-1 nude mice *n* = 6.

Group	Max TGI (%) ± SEM
HDM201 40 mg/kg qdx3	33.39 ± 13.90
HDM201 100 mg/kg qdx3	76.94 ± 5.38
Trametinib 0.3 mg/kg qdx6	65.47 ± 21.29
Trametinib 1 mg/kg qdx6	90.05 ± 1.13
HDM201 + Trametinib 40 + 0.3 mg/kg qdx3/qdx6	91.83 ± 1.37
HDM201 + Trametinib 40 + 1 mg/kg qdx3/qdx6	93.68 ± 1.63
HDM201 + Trametinib 100 + 0.3 mg/kg qdx3/qdx6	94.56 ± 1.77
HDM201 + Trametinib 100 + 1 mg/kg qdx3/qdx6	95.99 ± 0.84

**Table 8 ijms-23-12984-t008:** Summary of performed in vivo studies on CD-1 nude mice xenografted with A375 tumour.

Compound	Initial Tumour Volume (mm^3^)	Doses (mg/kg)	Dose Schedule	N	Comments
Vehicle	~162	-	qdx6	11	Efficacy
Siremadlin	~163–172	40/100	qdx3	6	Efficacy
Trametinib	~167–180	0.3/1	qdx6	6	Efficacy
Siremadlin+ Trametinib	~165–169	40 + 0.3/40 + 1/ 100 + 0.3/100 + 1	qdx3/qdx6	6	Efficacy
Siremadlin	~300	100	qdx1	12	PK
Trametinib	~300	1	qdx1	12	PK
Siremadlin+ Trametinib	~300	100 + 1	qdx1	12	PK

## Data Availability

The data presented in this study are available in the article or Appendix A. Raw data from MTS and RealTime-Glo assays and PK and PD studies are available on request from the corresponding author.

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
