# Peer review of "In Vitro/In Vivo Translation of Synergistic Combination of MDM2 and MEK Inhibitors in Melanoma Using PBPK/PD Modelling: Part I"

_ijms, 2022, doi:10.3390/ijms232112984_

Round 1

Reviewer 1 Report (Previous Reviewer 2)

Please see comments below regarding the authors’ responses in their carefully prepared cover letter and in corresponding sections of the revised manuscript:

 Point 1: The authors made an appropriate modification to their wording.

Point 2: This point remains unaddressed by the authors. If the delta value is vague, then please clarify through definition in the manuscript. It is uninformative as presented.

Point 3: The term and phrase in lines 111 and 127 are not equated in the text, so “translational PD parameter” is still undefined and largely meaningless for the reader. This must be corrected in the text.

Point 4: This was a meaningful modification by the authors.

Point 5: The inclusion of some PK data strengthens the paper. I applaud the authors for this effort.

Point 6: Just because it is difficult does not make it unimportant. The authors use in vitro models throughout the described study. There will be clear differences in combined drug performance between the in vitro and subsequent in vivo models employed. Does that make the in vitro model irrelevant? If so, then there is no reason for this work to be published (i.e., the model used is insufficient to mimic in vivo performance at any level). Similarly, although in vitro drug-resistant A375 melanoma cell cytotoxicity studies may not perfectly reflect the combination drug performance in drug-resistant tumors in animal models, a poor performance in an in vitro drug-resistant model would be valuable in showing the combination is highly unlikely to be suitable for treatment of such drug resistant cells in patients. I believe such drug performance data should be added to the described study for that important reason.

Point 7: This argument should be stated in the manuscript. Off target toxicity is still a possibility, particularly if the drug combination affects metabolism of one or both API, or their associated metabolites, but the clinical trial studies surely address such concerns.

Point 8: Could the negative control data be added to the cell viability plots in text? That is customary.

Point 9: This point should be reiterated for the reader in the Conclusion so they will be alerted to the associated PK-related manuscript (i.e., Part 2).

The authors made important and significant additions to the revised manuscript. Please see comments related to points 2, 3, 8, and 9 above for very simple additional changes that will enhance reader understanding. I also strongly suggest the authors consider point 6, although I recognize that may be cost or time prohibitive.

Author Response

Dear Reviewer,

We are very grateful for your valuable comments and advice. Please find our response in blue in the attached file.

Kind regards,

Jakub Witkowski

Reviewer 2 Report (New Reviewer)

The article entitled "In Vitro/In Vivo translation of synergistic combination of MDM2 and MEK inhibitors in melanoma using PBPK/PD modelling: Part I" by Jakub Witkowski has described the synergy between Siremadlin (MDM2 inhibitor) and Trametinib (MEK inhibitor) was performed In vitro cytotoxicity against A375 cancer cell lines. A detailed modelling study were carried out using Synergyfinder and Synergy packages and PK studies are also carried. 

The writeup in Pgae2 in introduction section from line 70-78 has to be concise and relevant Ref is to be provide.

The authors could have performed the efficacy of these combined drugs against the cancer targets. 

Some of the References has to be provide in method section particularly in vivo anticancer activity

Author Response

Dear Reviewer,

We are very grateful for your valuable comments and advice. Please find our response in blue in the attached file.

Kind regards,

Jakub Witkowski

This manuscript is a resubmission of an earlier submission. The following is a list of the peer review reports and author responses from that submission.

Round 1

Reviewer 1 Report

The subject of the manuscript is interesting and stimulates interest in the subject, however, the presented experimental material is surprisingly scarce. One A375 cell line was selected for the study, and many more are known (Mol Cancer Ther. 2009 May; 8 (5): 1292–1304. Doi: 10.1158 / 1535-7163.MCT-08-1030).

Why was only one A375 cell line selected?

Why were cells incubated for 72 hours?

How can you explain that in Figures 1 and 2, the viability of cells increases with incubation of 12 and 24 hours and decreases with increasing time? Is there a necrosis?

There are too few experiments in the manuscript described in relation to theoretical research that are less valuable without detailed experimental research. In truth, the authors mentioned the continuation of these studies, but there are very few results in this manuscript to be positive about the combination of these drugs and their use in the treatment of malignant melanoma. Instead of mentioning that "further PK / PD studies on mice coupled with PBPK / PD modeling are needed in order to select the optimal PD interaction parameter which will translate observed in vitro synergy metrics into the in vivo settings", this manuscript should be supplemented with these results. Before publication, the manuscript needs to be supplemented with the results of more detailed experimental studies (e.g. using several melanoma lines, experiments on zebra fish or mice). How does this drug combination affect the relevant signaling pathways? Completely overlooked aspect.

In this version, I do not recommend the manuscript for publication.

Author Response

Dear Reviewer,

We would like to thank you for your valuable comments and your in-depth analysis of our manuscript. Please find our point-by-point response down below and also please see the attachment with the second co-submitted manuscript.

Point 1: Why was only one A375 cell line selected?

Response 1: That is a very good question, thank you. Indeed there are many more established malignant melanoma cell lines however for not every one of them it is possible to obtain xenograft in mice. In fact panel of tested malignant melanoma cell lines and drug combinations was broader (data not shown). Among tested malignant melanoma cell lines (A375, HT144, SKMEL5) A375 cell line was selected due to:

  • Ability to form xenograft and relatively fast-growing tumour mass in mice
  • Proven efficacy of MDM2 and MEK inhibitors both in vitro/in vivo (doi: 3390/cancers11010003 and WO2015070224)
  • A375 cells are the informal golden standard in testing drugs targeting MAPK/ERK pathway which is reflected in the highest number of PubMed citations (doi:18632/oncotarget.14443)

Point 2: Why were cells incubated for 72 hours?

Response 2: We very much appreciate this question, thank you. Cells were incubated for 72 hours because:

  • This time allowed to not overgrow (reach 100% of confluence) of negative control with DMSO vehicle and to fully deplete nutrients from DMEM medium which could affect obtained cytotoxicity data.
  • 72 hours of incubation was an optimal time to obtain reliable dose-response data in MTS and RealTime-Glo assays
  • RealTimeGlo assay is designed for up to 72 hours of incubation as described in assay protocol (luminescent signal is decreasing after 72h of incubation which is the most significant for high seeding of cells).

Point 3: How can you explain that in Figures 1 and 2, the viability of cells increases with incubation of 12 and 24 hours and decreases with increasing time? Is there a necrosis?

Response 3: Thank you for this excellent question. From our previous studies (data not shown) we know that those drugs are causing apoptosis or cell cycle arrest depending on which stage of cell cycle were particular cells which is in line with published literature doi: 10.1073/pnas.050749310. Regarding the increase of viability after short incubation time, it may be related to increased mitochondrial activity of the cells (this may cause increase NanoLuc enzyme turnover in RealtimeGlo assay) which was described for some cytotoxic drugs shortly incubated with cells e.g. 8 and 24h (please see doi: 10.3390/molecules27092693).

Point 4: There are too few experiments in the manuscript described in relation to theoretical research that are less valuable without detailed experimental research. In truth, the authors mentioned the continuation of these studies, but there are very few results in this manuscript to be positive about the combination of these drugs and their use in the treatment of malignant melanoma. Instead of mentioning that "further PK / PD studies on mice coupled with PBPK / PD modeling are needed in order to select the optimal PD interaction parameter which will translate observed in vitro synergy metrics into the in vivo settings", this manuscript should be supplemented with these results. Before publication, the manuscript needs to be supplemented with the results of more detailed experimental studies (e.g. using several melanoma lines, experiments on zebra fish or mice). How does this drug combination affect the relevant signaling pathways? Completely overlooked aspect.

Response 4: Thank you for your valuable comments and your in-depth analysis of our manuscript. We agree that this manuscript can make an impression of presenting too few experimental results. However only apparently because the main goal of this manuscript was not to investigate this drug combination in depth at the level of effect on particular signaling pathways, but to investigate features of in vitro data that could be used in further PBPK/PD modelling on in vivo data. Moreover in the original publication plan results from current in vitro studies were intended to be presented with in vivo PK/PD and PBPK/PD modelling data. Due to the large amount of obtained results and the length of the original manuscript which was over 50 pages and could be uneasy to read, we decided to divide it into a cycle of 3 manuscripts. As mentioned in the cover letter results presented in this manuscript are only an introduction to the topic of in vitro/in vivo translation of drug combination data and are strongly connected to the second manuscript. An exemplary MEK/MDM2 combination was selected due to availability of preclinical and clinical PK/PD data and established/confirmed molecular interaction mechanism as well. We hope that attached copy of second part of this publication cycle (which is also submitted to the IJMS with manuscript ID: ijms-1906160) including mentioned above following in vivo PK/PD and PBPK/PD modelling data for this drug combination will dispel doubts related to the amount of the presented results.
It is also worth mentioning that a very valuable added value of this manuscript is the R and Python code allowing for calculation of various drug interaction metrics from data from high throughput screening which was also used to analyze data from over 400 already performed experiments (data not shown).

Reviewer 2 Report

The authors investigated possible synergism between Siremadlin and Trametinib in A375 melanoma cells using MTS and RealTime-GLO cytotoxicity assays, with evaluation of interaction metrics using Synergy and Synergyfinder software. A375 cells exhibited both dose- and time-dependent cell death responses to the drug combination. The authors claim the positive results from these A375 melanoma cell cytotoxicity studies warrant further evaluation in murine models.

The following statements in the manuscript require attention:

·         [lines 53-56] The authors state: “Bench to bedside approach for drug combination is much more challenging than for monotherapy because it must account for the interaction between two (or even more) drugs at two different levels: Pharmacokinetic (PK) and Pharmacodynamic (PD).” I firmly disagree with this statement. The statement may be true relative to repurposing monotherapies, but certainly is not true relative to novel monotherapy drug candidates in which no data regarding PD/PK is yet established in higher mammals or humans. At least with combination therapies involving established drugs or those in active clinical trials—as considered herein--the safety, efficacy, and metabolic profile of each drug is already at least partially established. That is a major advantage versus the overwhelming majority of early drug candidate monotherapies.

·         [lines 97-99] The statement: “For example, a delta of 10 would indicate that the drug combination will produce on average 10% more response compared to the expected effect which we would refer here as synergistic drugs interaction, while a delta of −10 would indicate an antagonistic drugs interaction with the same level of magnitude for this case” is unclear. 10% more response compared to what--the additive effect of the drugs?

·         [line 111] “translation PD parameter” is not defined by the authors, yet it is the focus of the question to be answered through the investigation.

·         [lines 174] “however, it seems that synergistic cooperativity is more important in neurological disorders than in treating cancer, thus lack of synergy in this metric is negligible.” The authors’ point here is unclear and should be restated for clarity. Do they mean ‘clinically inconsequential’ rather than “negligible” or am I misinterpreting their intended meaning?

Additionally:

·         In addition, the authors repeatedly mention potential translational application and necessary PK studies of combination therapies, but no PK data of any type were generated with the selected drug combination.

·         As noted by the authors, an advantage to combination therapies is the possibility to circumvent resistant mechanisms associated with at least one of the combined drugs, yet the authors did not evaluate the drug combination in drug-resistant strains of A375 melanoma cells.

·         Is the drug combination cytotoxic in healthy cells over the extended durations?

·         Why were negative controls (e.g., vehicle) not employed, or at least not plotted, in the cell viability assays?   

In summary, the level of demonstrated enhanced efficacy in treatment of A375 melanoma cells with Siremadlin and Trametinib as described herein is modest at best. Considering many thousands of drug combinations are routinely evaluated in low- and high-throughput assays against all manner of cells, I do not find the study or the results compelling. The authors emphasize the potential significance of the combination therapy but do little to demonstrate it. Even so, the preliminary results are positive, albeit uninspiring, so I hesitantly recommend publication.

Author Response

Dear Reviewer,

We would like to thank you for your valuable comments and your in-depth analysis of our manuscript. Please find our point-by-point response down below and also please see the attachment with the second co-submitted manuscript.

Point 1: [lines 53-56] The authors state: “Bench to bedside approach for drug combination is much more challenging than for monotherapy because it must account for the interaction between two (or even more) drugs at two different levels: Pharmacokinetic (PK) and Pharmacodynamic (PD).” I firmly disagree with this statement. The statement may be true relative to repurposing monotherapies, but certainly is not true relative to novel monotherapy drug candidates in which no data regarding PD/PK is yet established in higher mammals or humans. At least with combination therapies involving established drugs or those in active clinical trials—as considered herein--the safety, efficacy, and metabolic profile of each drug is already at least partially established. That is a major advantage versus the overwhelming majority of early drug candidate monotherapies.

Response 1: Thank you very much for this valuable comment. A combination consisting of MDM2 and MEK inhibitors is the subject of many studies in the clinical setting (ClinicalTrials.gov identifiers: NCT02110355, NCT03714958, NCT02016729, NCT01985191, NCT03566485). This drug combination utility was confirmed in the clinical setting with moderately active MDM2 inhibitor AMG232 (doi: 10.1200/JCO.2017.35.15_suppl.2575) but it is believed that the next generation of more potent MDM2 inhibitors like Siremadlin (HDM201) may enhance this synergistic drug interaction. The proposed combination of Siremadlin (currently Phase 2 according to Clinicaltrials.gov) and Trametinib (already approved therapy) is already undergoing clinical examination (NCT03714958, results not published yet) thus it is believed that concerns related to the safety, efficacy and metabolic profile of each drug are already at least partially established. Due to the complexity of cancer treatment at almost every stage of anticancer drug development design of drug combinations are crucial. When the safety and efficacy of monotherapy are proven then a successful drug combination approach is increasing probability of the therapy approval. Furthermore, the initial drug combination data showing improved efficacy are required by regulatory agencies to start a clinical investigation.
Regarding bench to bedside process, in a revised version of the manuscript this sentence was changed to “Bench to bedside approach for drug combination may be possible only when PK/PD for both drugs data is available because it must account for the interaction between two (or even more) drugs at two different levels: Pharmacokinetic (PK) and Pharmacodynamic (PD).”

Point 2: [lines 97-99] The statement: “For example, a delta of 10 would indicate that the drug combination will produce on average 10% more response compared to the expected effect which we would refer here as synergistic drugs interaction, while a delta of −10 would indicate an antagonistic drugs interaction with the same level of magnitude for this case” is unclear. 10% more response compared to what--the additive effect of the drugs?

Response 2: Thank you for a great question. The definition of the expected effect value of drug combination is vague in the literature. However, it must be clear that each of the models defines the expected value in a different way depending on the assumptions of the given model which is greatly described and visualized by Foucquier and Guedj (see doi:10.1002/prp2.149, Figure 1). Since Synergyfinder is utilizing different drug interaction models (Loewe, Bliss, HSA, ZIP) thus delta score must be universal in terms of expected value (additive effect) metric.

Point 3: [line 111] “translation PD parameter” is not defined by the authors, yet it is the focus of the question to be answered through the investigation.

Response 3: Thank you for noticing that. In line 111 it is mentioned what drug interaction metrics could be used as ‘’translation PD parameter’’ which is further explained in line 127: ‘’PD interaction parameter which may serve for translatability of in vitro drug combination results into in vivo settings.’’

Point 4: [line 174] “however, it seems that synergistic cooperativity is more important in neurological disorders than in treating cancer, thus lack of synergy in this metric is negligible.” The authors’ point here is unclear and should be restated for clarity. Do they mean ‘clinically inconsequential’ rather than “negligible” or am I misinterpreting their intended meaning?

Response 4: That is a good point, thank you. In the corrected version of the manuscript this sentence was changed to be more clear: ‘’ however, it seems that synergistic cooperativity is more important in neurological disorders than in treating cancer, thus lack of synergy in this metric is clinically not relevant.’’

Point 5: In addition, the authors repeatedly mention potential translational application and necessary PK studies of combination therapies, but no PK data of any type were generated with the selected drug combination.

Response 5: That is another good point however in the original publication plan results from current in vitro studies were intended to be presented with in vivo PK/PD and PBPK/PD modelling data. Due to the large amount of obtained results and the length of the original manuscript which was over
50 pages and could be uneasy to read, we decided to divide it into a cycle of 3 manuscripts. All PK and PD data for this drug combination is reported in the second part of publication cycle. This manuscript is strictly related to the determination of PD interaction parameters and investigation of in vitro data features that could be used in further PBPK/PD modelling on in vivo data.

Point 6: As noted by the authors, an advantage to combination therapies is the possibility to circumvent resistant mechanisms associated with at least one of the combined drugs, yet the authors did not evaluate the drug combination in drug-resistant strains of A375 melanoma cells.

Response 6: Thank you for asking for that. We agree that further evaluation in drug-resistant A375 sublines might be useful however according to the results presented by Hoffman-Luca et al. (see doi: 10.1371/journal.pone.0128807) there are differences in acquired resistance between in vitro and in vivo models thus it might be difficult to employ such data for further PBPK/PD modelling and simulation.

Point 7: Is the drug combination cytotoxic in healthy cells over the extended durations?

Response 7: That is a very good and fundamental question which should be asked about every drug combination. Examination of this particular drug combination on healthy cells was not in the scope of this publication but based on previously published data for MDM2 inhibitors on healthy cells (doi: 10.1073/pnas.0708917105 it is known that MDM2 inhibitors are transiently inducing cell cycle arrest but without inducing PUMA which is a potent mediator of p53-dependent apoptosis. Moreover, since this drug combination is already under clinical evaluation it is assumed that this combination is safe and not cytotoxic to healthy cells.

Point 8: Why were negative controls (e.g., vehicle) not employed, or at least not plotted, in the cell viability assays?  

Response 8: Thank you for this question. Negative controls (Control DMSO) were used in both MTS and RealTime-Glo assays as shown in supplementary Figures S1 and S2.

Point 9: In summary, the level of demonstrated enhanced efficacy in treatment of A375 melanoma cells with Siremadlin and Trametinib as described herein is modest at best. Considering many thousands of drug combinations are routinely evaluated in low- and high-throughput assays against all manner of cells, I do not find the study or the results compelling. The authors emphasize the potential significance of the combination therapy but do little to demonstrate it.

Response 9: We very much appreciate this comment. It should be noted that A375 cells can be classified as an aggressive phenotype melanoma cells (see doi: 10.3390/cancers12103018 and doi: 10.1186/s12964-022-00871-x) thus therapeutic outcome of monotherapy and drug combination may not be spectacular. Nevertheless, we must be aware that observed synergistic drug interaction may be further enhanced in vivo by interaction at the PK level which is discussed in the second part of the publication cycle. As mentioned in the cover letter results presented in this manuscript are only
an introduction to the topic of in vitro/in vivo translation of drug combination data and are strongly connected to the second manuscript. We hope that attached copy of second part of this publication cycle (which is also submitted to the IJMS with manuscript ID: ijms-1906160) including following
in vivo PK/PD and PBPK/PD modelling data for this drug combination will dispel any doubts related to publication process of this manuscript.
